# Improving Intrinsic Exploration with Language Abstractions

**Jesse Mu[1]\*, Victor Zhong[2,3], Roberta Raileanu[3], Minqi Jiang[3,4],**
**Noah Goodman[1], Tim Rocktäschel[4]\*, Edward Grefenstette[4,5]\***

[1]Stanford University, [2]University of Washington, [3]Meta AI, [4]University College London, [5]Cohere

## Abstract

Reinforcement learning (RL) agents are particularly hard to train when rewards are sparse. One common solution is to use *intrinsic* rewards to encourage agents to explore their environment. However, recent intrinsic exploration methods often use state-based novelty measures which reward low-level exploration and may not scale to domains requiring more abstract skills. Instead, we explore *language* as a general medium for highlighting relevant abstractions in an environment. Unlike previous work, we evaluate whether language can improve over existing exploration methods by directly extending (and comparing to) competitive intrinsic exploration baselines: AMIGo (Campero et al., 2021) and NovelD (Zhang et al., 2021). These language-based variants outperform their non-linguistic forms by 47–85% across 13 challenging tasks from the MiniGrid and MiniHack environment suites.

## 1 Introduction

A central challenge in reinforcement learning (RL) is designing agents that can solve complex, long-horizon tasks with sparse rewards. In the absence of extrinsic rewards, one popular solution is to provide *intrinsic* rewards for exploration [34, 35, 42, 43]. This invariably leads to the challenging question: how should one measure exploration? One common answer is that an agent should be rewarded for attaining "novel" states in the environment, but naive measures of novelty have limitations. For example, consider an agent that starts in the kitchen of a large house and must make an omelet. Simple state-based exploration will reward an agent for visiting every room in the house, but a more effective strategy would be to stay put and use the stove. Moreover, like kitchens with different-colored appliances, states can look cosmetically different but have the same underlying semantics, and thus are not truly novel. Together, these constitute two fundamental challenges for intrinsic exploration: first, how can we reward true progress in the environment over meaningless exploration? Second, how can we tell when a state is not just superficially, but *semantically* novel?

Fortunately, humans are equipped with a powerful tool for solving both problems: language. As a cornerstone of human intelligence, language has strong priors over the features and behaviors needed for exploration and skill acquisition. It also describes a rich and compositional set of meaningful behaviors as simple as directions (e.g. *move left*) and as abstract as conjunctions of high level tasks (e.g. *retrieve the ring and defeat the wizard*) that can categorize and unify many possible world states.

Our aim is to see whether language abstractions can improve existing state-based exploration methods in RL. While language-guided exploration methods exist in the literature [3, 5, 12, 13, 21–24, 31, 44, 51, 53], we make two key contributions over prior work. First, existing methods assume access to a high-level linguistic instruction for reward shaping, or otherwise assume that any intermediate language annotations encountered are always helpful for learning. Instead, we study settings without

---

\*Work done while at Meta AI. Correspondence to `muj@stanford.edu`

36th Conference on Neural Information Processing Systems (NeurIPS 2022).

instructions, with more diverse intermediate messages (Figure 1) that may or may not be useful, but may nonetheless be a more effective measure of novelty than raw states.

Second, past work often compares only to vanilla RL, while ignoring competitive intrinsic exploration baselines. This leaves the true utility of language over simpler state-based exploration unclear. To remedy this issue, we conduct a controlled evaluation on the effect of language on competitive approaches to exploration by extending two recent, state-of-the-art methods: AMIGo [7], where a teacher proposes intermediate location-based goals for a student, and NovelD [54], which rewards an agent for visiting novel regions of the state space. Building upon these methods, we propose **L-AMIGo**, where the teacher proposes goals expressed via language instead of coordinates, and **L-NovelD**, a variant of NovelD with an additional exploration bonus for visiting linguistically-novel states.

Across 13 challenging, procedurally-generated, sparse-reward tasks in the MiniGrid [8] and MiniHack [41] environment suites, we show that language-parameterized exploration methods outperform their non-linguistic counterparts by 47–85%, especially in more abstract tasks with larger state and action spaces. We also show that language improves the interpretability of the training process, either by developing a natural curriculum of semantic goals (in L-AMIGo) or by allowing us to visualize the most novel language during training (in L-NovelD). Finally, we show when and where the fine-grained compositional semantics of the language improves agent exploration, when compared to non-compositional baselines.

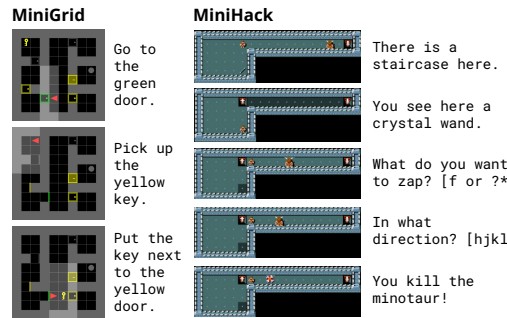

Figure 1: **Language conveys meaningful environment abstractions.** Language state annotations in the MiniGrid KeyCorridorS4R3 [8] and MiniHack Wand of Death (Hard) [41] tasks.

## 2   Related Work

**Exploration in RL.**   Exploration has a long history in RL, from $\varepsilon$-greedy [48] or count-based exploration [4, 29, 30, 32, 47, 50] to intrinsic motivation [33–35] and curiosity-based learning [42]. More recently, deep neural networks have been used to measure novelty with changes in state representations [6, 40, 54] or prediction errors in world models [1, 36, 46]. Another long tradition generates curricula of intrinsic goals to encourage learning [7, 13, 14, 18–20, 37–39]. In this paper, we explore the potential benefit of language on these approaches to exploration.

**Language for Exploration.**   The observation that language-guided exploration can improve RL is not new: language has been used to shape policies [24, 51] and rewards [3, 5, 21–23, 31, 44, 53] and set intrinsic goals [12, 13]. Crucially, our work differs from prior work in two ways: first, instead of reward shaping with high-level instructions, we use noisier, intermediate language annotations for exploration; second, we directly extend and compare to competitive intrinsic exploration baselines.

L-AMIGo, our variant of AMIGo with language goals, is similar to the IMAGINE agent of Colas et al. [13], which also sets intrinsic language goals. However, IMAGINE is built for instruction following, and requires a perfectly compositional space of language goals, which the agent tries to explore so that it can complete novel goals at test time. Instead, we make no assumptions on the language and explore to maximize extrinsic reward, using an alternative *goal difficulty* metric to measure progress.

Meanwhile, reward shaping and inverse RL methods [3, 5, 21–24, 31, 44, 51, 53] reward an agent for actions associated with linguistic descriptions, but again are primarily designed for instruction following, where an extrinsic goal is available to help shape intermediate rewards. In our setting, however, we have not high-level extrinsic goals but low-level *intermediate* language annotations. Extrinsic reward shaping methods such as LEARN [22] could be naively applied by simply doing reward shaping with every intermediate language annotation, and a few of these methods are designed for low-level language subgoals [24, 31]. However, a shared assumption of these approaches is that *language is always helpful*: either because we have expert-curated messages (as in Harrison et al. [24]), or because we have goal descriptions that let us identify subgoals relevant to the extrinsic goal (as in ELLA; Mirchandani et al. [31]). In our tasks, however, most language is *unhelpful* for progress in the environment, and we have no extrinsic goals. Consequently, past methods reduce to

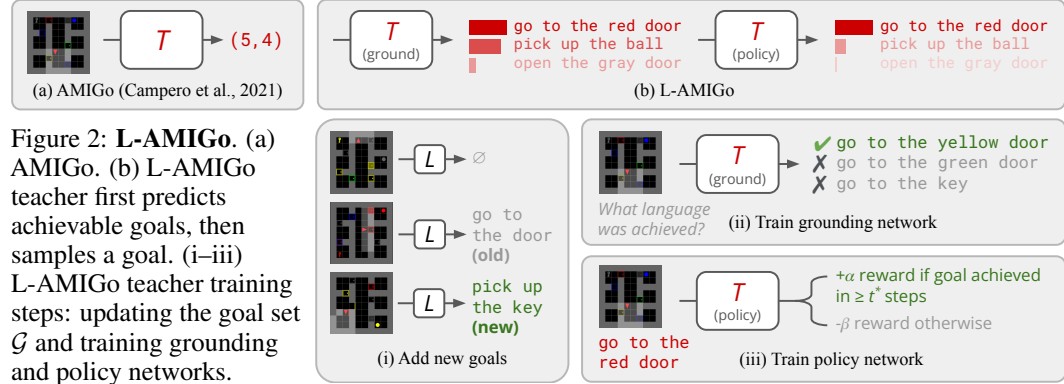

Figure 2: **L-AMIGo**. (a) AMIGo. (b) L-AMIGo teacher first predicts achievable goals, then samples a goal. (i–iii) L-AMIGo teacher training steps: updating the goal set $\mathcal{G}$ and training grounding and policy networks.

simply giving a fixed reward for every intermediate message encountered, which (we will show) fails to learn. Finally, work concurrent to ours by Tam et al. [49] tackles similar ideas in photorealistic environments that permit transfer from foundation models. Instead, we explore domain-specific symbolic games with built-in language where such models are not readily available.

A final distinguishing contribution of our work is that prior work often neglects non-linguistic exploration baselines. For example, Harrison et al. [24] and LEARN [22] compare to vanilla RL only; ELLA [31] compares to LEARN and RIDE [40], with limited improvements over RIDE. Prior work can thus be summarized as showing that linguistic rewards improve over extrinsic rewards alone. Instead, we provide novel evidence that *linguistic rewards improve upon state-based intrinsic rewards*, using the same exploration methods and challenging tasks typical of recent work in RL.

## 3  Problem Statement

We explore RL in the setting of an augmented Markov Decision Process (MDP) defined as the tuple $(\mathcal{S}, \mathcal{A}, T, R, \gamma, L)$, where $\mathcal{S}$ and $\mathcal{A}$ are the state and action spaces, $T : \mathcal{S} \times \mathcal{A} \to \mathcal{S}$ is the environment transition dynamics, $R : \mathcal{S} \times \mathcal{A} \to \mathbb{R}$ is the *extrinsic* reward function, where $r_t = R(s_t, a_t)$ is the reward obtained at time $t$ by taking action $a_t$ in state $s_t$, and $\gamma$ is the discount factor. To add language, we assume access to an *annotator* $L$ that produces **language descriptions** for states: $\ell_t = L(s_t)$, such as those in Figure 1. Note that not every state needs a description (which we model with a null description $\varnothing$) and the set of descriptions need not be known ahead of time.[2] We ultimately seek a policy that maximizes the expected discounted (extrinsic) reward $R_t = \mathbb{E}[\sum_{k=0}^{H} \gamma^k r_{t+k}]$, where $H$ is the finite time horizon. During training, however, we maximize an *augmented* reward $r_t^+ = r_t + \lambda r_t^i$, where $r_t^i$ is an *intrinsic* reward and $\lambda$ is a scaling hyperparameter.

Like past work [26, 31, 53] we make the simplifying assumption of access to an oracle language annotator $L$ provided by the environment. Note that the language annotator is "oracle" in that it always outputs messages that are true of the current state, but *not* "oracle" in that it indiscriminately outputs messages that are not necessarily relevant for the extrinsic goal. Many modern RL environments are pre-equipped with language, including NetHack/MiniHack [28, 41], text-based games [15, 45, 52], and in fact most video games in general. In the absence of an oracle annotator, one common approach is to learn an annotator model from a dataset of language-annotated states [3, 22, 31], though such datasets are often generated from oracles that are simply run offline instead [31]. Concurrent work [49] uses pretrained foundation models to automatically provide annotations in 3D environments, though such models are not readily available in the symbolic 2D games we explore. Since this idea has been well-proven, we assume oracle access to $L$, but as an example, one could straightforwardly adapt the annotator model trained on BabyAI by Mirchandani et al. [31] to our setting.

---

[2]For presentational simplicity, the annotator here outputs a single description per state, but in practice, we allow an annotator to produce *multiple* descriptions: e.g. in MiniGrid, *open the door* and *open the red door* describe the same state. This requires two minor changes in the equations, described in Footnotes 4 and 5.

# 4 L-AMIGo

We now describe our approach to jointly training a student and a goal-proposing teacher, extending AMIGo [7] to arbitrary language goals.

## 4.1 Adversarially Motivated Intrinsic Goals (AMIGo)

AMIGo [7] augments an RL student policy with goals generated by a teacher, which provide intrinsic reward when completed (Figure 2a). The idea is that the teacher should propose intermediate goals that start simple, but grow harder to encourage an agent to explore its environment.

**Student.** Formally, the student $S$ is a *goal-conditioned* policy parameterized as $\pi_S(a_t \mid s_t, g_t; \theta_S)$, where $g_t$ is the goal provided by the teacher, and the student receives an intrinsic reward $r_t^i$ of 1 only if the teacher's goal at that timestep is completed. The student receives a goal from the teacher either at the beginning of an episode, or mid-episode, if the previous goal has been completed.

**Teacher.** Separately, AMIGo trains an adversarial *teacher* policy $\pi_T(g_t \mid s_0; \theta_T)$ to propose goals to the student given its initial state. The teacher is trained with a reward $r_t^T$ that depends on a *difficulty threshold* $t^*$: the teacher is given a positive reward of $+\alpha$ for proposing goals that take the student more than $t^*$ timesteps to complete, and $-\beta$ for goals that are completed sooner, or never completed within the finite time horizon. To encourage proposing harder and harder goals that promote exploration, $t^*$ is increased linearly throughout training: whenever the student completes 10 goals in a row under the current difficulty threshold, it is increased by 1, up to some tunable maximum difficulty. Finally, to encourage intermediate goals that are aligned with the extrinsic goal, the teacher is also rewarded with the extrinsic reward when the student attains it.

This teacher is updated separately from the student at different time intervals. Formally, its training data is batches of $(s_0, g_t, r_t^T)$ tuples collected from student trajectories for nonzero $r_t^T$, where $s_0$ is the initial state of the student's trajectory and $g_t$ is the goal that led to reward $r_t^T$.

The original paper [7] implements AMIGo for MiniGrid only, where the goals $g_t$ are $(x, y)$ coordinates to be reached. The student gets the goal embedded directly in the $M \times N$ environment, and the teacher is a dimensionality-preserving convolutional network which encodes the student's $M \times N$ environment into an $M \times N$ distribution over coordinates, from which a single goal is selected.

## 4.2 Extension to L-AMIGo

**Student.** The L-AMIGo student is a policy conditioned not on $(x, y)$ goals, but on *language goals* $\ell_t$: $\pi_S(a_t \mid s_t, \ell_t; \theta_S)$. Given the "goal" $\ell_t$, the student is now rewarded if it reaches a state with the language description $\ell_t$, i.e. if $\ell_t = L(s_t)$.[3] Typically this student will encode the goal with a learned language model and concatenate the goal representation with its state representation.

**Teacher.** Now the L-AMIGo teacher selects goals from the set of possible language descriptions in the environment. Because the possible goals are initially unknown, the teacher maintains a running set of goals $\mathcal{G}$ that is updated as the student encounters new state descriptions (Figure 2b).

This move to language creates a challenge: not only must a teacher choose a goal to propose, it must also determine which goals are achievable at all. For example, the goal *go to the red door* only makes sense in environments with red doors. In L-AMIGo, these tasks are factorized into a **policy network**, which produces the distribution over goals given a student's state, and a **grounding network**, which predicts the probability that a goal is likely to be achieved in the first place (Figure 2b):

$$\pi_T(\ell_t \mid s_t; \theta_T) \propto p_{\text{ground}}(\ell_t \mid s_t; \theta_T) \cdot p_{\text{policy}}(\ell_t \mid s_t; \theta_T) \tag{1}$$

$$p_{\text{ground}}(\ell_t \mid s_t; \theta_T) = \sigma\left(f(\ell_t; \theta_T) \cdot h_{\text{ground}}(s_t; \theta_T)\right) \tag{2}$$

$$p_{\text{policy}}(\ell_t \mid s_t; \theta_T) \propto f(\ell_t; \theta_T) \cdot h_{\text{policy}}(s_t; \theta_T) \tag{3}$$

---

[3]We can treat language *goals* and *state descriptions* equivalently, even if the wordings are slightly different across environments. In MiniGrid, messages (e.g. *go to the red door*) look like goals but can also be interpreted as state descriptions: *[in this state, you have] go[ne] to the red door* In MiniHack, messages are description-like (e.g. *you kill the minotaur!*), but imagine the teacher's goal as *[reach a state where] you kill the minotaur!*

Equation 3 describes the policy network as producing a probability for a goal by computing the dot product between goal and state representations $f(\ell_t; \theta_T)$ and $h_{\text{policy}}(s_t; \theta_T)$, normalizing over possible goals; this policy is learned identically to the standard AMIGo teacher (Figure 2iii). Equation 2 specifies the grounding network as predicting whether a goal is *achievable* in an environment, by applying the sigmoid function to the dot product between the goal representation $f(\ell_t; \theta_T)$ and a (possibly separate) state representation $h_{\text{ground}}(s_t; \theta_T)$. Given an oracle grounding classifier, which outputs only 0 or 1, this is equivalent to restricting the teacher to proposing only goals that are achievable in a given environment. In practice, however, we learn the classifier online (Figure 2i). Given the initial state $s_0$ of an episode, we ask the grounding network to predict the first language description encountered along this trajectory: $\ell_{\text{1st}} = L(s_{t'})$, where $t'$ is the minimum $t$ where $L(s_t) \neq \varnothing$. This is formalized as a multilabel binary cross entropy loss,

$$\mathcal{L}_{\text{ground}}(s_0, \ell_{\text{1st}}) = -\log(p_{\text{ground}}(\ell_{\text{1st}} \mid s_0; \theta_T)) - \tfrac{1}{|\mathcal{G}|-1} \sum_{\ell' \in \mathcal{G} \setminus \{\ell_{\text{1st}}\}} \log(1 - p_{\text{ground}}(\ell' \mid s_0; \theta_T)), \quad (4)$$

where the second term noisily generates negative samples of (start state, unachieved description) pairs based on the set of descriptions $\mathcal{G}$ known to the teacher at the time, similar to contrastive learning.[4] Note that since $\mathcal{G}$ is updated during training, Equation 4 grows to include more terms over time.

To summarize, training the teacher involves three steps: (1) updating the running set of descriptions seen in the environment, (2) learning the policy network based on whether the student achieved goals proposed by the teacher, and (3) learning the grounding network by predicting descriptions encountered from initial states. Algorithm S1 in Appendix A describes how L-AMIGo trains in an asynchronous actor-critic framework, where the student and teacher are jointly trained from batches of experience collected from separate actor threads, as used in our experiments (see Section 6).

## 5 L-NovelD

Next, we describe NovelD [54], which extends simpler tabular- [47] or pseudo- [4, 6] count-based intrinsic exploration methods, and our language variant, L-NovelD. Instead of simply rewarding an agent for rare states, NovelD rewards agents for *transitioning* from states with low novelty to states with higher novelty. Zhang et al. [54] show that NovelD surpasses Random Network Distillation [6], another popular exploration method, on a variety of tasks including MiniGrid and Atari.

### 5.1 NovelD

NovelD defines the reward $r_t^i$ to be the difference in novelty between state $s_t$ and previous state $s_{t-1}$:

$$r_t^i = \text{NovelD}_s(s_t, s_{t-1}) \triangleq \underbrace{\max(N(s_t) - \alpha N(s_{t-1}), 0)}_{\text{Term 1 (NovelD)}} \cdot \underbrace{\mathbb{1}(N_e(s_t) = 1)}_{\text{Term 2 (ERIR)}}. \quad (5)$$

In the first NovelD term, $N(s_t)$ is the novelty of state $s_t$; this quantity describes the difference in novelty between successive states, which is clipped $> 0$ so the agent is not penalized from moving back to less novel states. $\alpha$ is a hyperparameter that scales the average magnitude of the reward. The second term is the **E**pisodic **R**eduction on **I**ntrinsic **R**eward (ERIR): a constraint that the agent only receives reward when encountering a state *for the first time in an episode*. $N_e(s_t)$ is an episodic state counter that tracks exact state visitation counts, as defined by $(x, y)$ coordinates.

**Measuring novelty with RND.** In smaller MDPs, it is possible to track exact state visitation counts, in which case the novelty is typically the inverse square root of visitation counts [47]. However, in larger environments where states are rarely revisited, we (like NovelD) use the popular Random Network Distillation (RND) [6] technique as an approximate novelty measure. Specifically, the novelty of a state is measured by the prediction error of a state embedding network that is trained jointly with the agent to match the output of a fixed, random target network. The intuition is that states which the RND network has been trained on will have lower prediction error than novel states.

---

[4]If multiple "first" descriptions are found, the teacher predicts 1 for *each* description, and 0 for all others.

## 5.2 Extension to L-NovelD

Our incorporation of language is simple: we add an additional exploration bonus based on novelty defined according to the language descriptions of states:

$$\text{NovelD}_\ell(\ell_t, \ell_{t-1}) \triangleq \max(N(\ell_t) - \alpha N(\ell_{t-1}), 0) \cdot \mathbb{1}(N_e(\ell_t) = 1). \tag{6}$$

This bonus is identical to standard NovelD: $N(\ell)$ is the novelty of the description $\ell$ as measured by a *separately parameterized* RND network encoding the description,[5] and $N_e(\ell_t) = 1$ when the language description has been encountered for the first time this episode. We keep the original NovelD exploration bonus, as language rewards may be sparse and a basic navigation bonus can encourage the agent to reach language-annotated states. The final intrinsic reward for L-NovelD is

$$r_t^i = \text{L-NovelD}(s_t, s_{t-1}, \ell_t, \ell_{t-1}) \triangleq \text{NovelD}_s(s_t, s_{t-1}) + \lambda_\ell \text{NovelD}_\ell(\ell_t, \ell_{t-1}) \tag{7}$$

where $\lambda_\ell$ controls the trade-off between Equations 5 and 6.

One might ask why we do not simply include the language description $\ell$ as input into the RND network, along with the state. While this can work in some cases, decoupling the state and language novelties allow us to precisely control the trade-off between the two, with a hyperparameter that can be tuned to different tasks. In contrast, a combined input obfuscates the relative contributions of state and language to the overall novelty. Appendix F.2 has ablations that show that (1) combining the state and language inputs or (2) using the language novelty term alone leads to worse performance.

## 6 Experiments

We evaluate L-AMIGo, AMIGo, L-NovelD, and NovelD, implemented in the TorchBeast [27] implementation of IMPALA [17], a common asynchronous actor-critic method. Besides vanilla IMPALA, we also compare to a naive (fixed) message reward given for any message in the environment, which is similar doing extrinsic reward shaping for all messages (e.g. LEARN [22]; also [3, 5, 21, 23, 44, 53]) or prior approaches that assume that messages are always helpful (Harrison et al. [24], ELLA [31]); see Appendix B for more discussion on this baseline and its equivalencies to prior work.[6] We run each model 5 times across 13 tasks within two challenging procedurally-generated RL environments, MiniGrid [8] and MiniHack [41], and adapt baseline models provided for both environments [7, 41]; for full model, training, and hyperparameter details, see Appendix C.

### 6.1 Environments

**MiniGrid.** Following Campero et al. [7], we evaluate on the most challenging tasks in MiniGrid [8], which involve navigation and manipulation tasks in gridworlds: **KeyCorridorS{3,4,5}R3** (Figure 1) and **ObstructedMaze_{1Dl,2Dlhb,1Q}**. These tasks involve picking up a ball in a locked room, with the key to the door hidden in boxes or other rooms and the door possibly obstructed. The suffix indicates the size of the environment, in increasing order. See Appendix G.1 for more details.

To add language, we use the complementary BabyAI platform [9] which provides a grammar of 652 possible messages, involving *goto*, *open*, *pickup*, and *putnext* commands applied to a variety of objects qualified by type (e.g. *box*, *door*) and/or color (e.g. *red*, *blue*). The oracle language annotator emits a message when the corresponding action is completed. On average, only 6 to 12 messages (1-2% of all 652) are needed to complete each task (see Appendix G.1 for all messages).

Note that since BabyAI messages are not included in the original environment from which we adapt baseline agents [7], none of our MiniGrid agents encode language observations directly into the state. While it can be tempting and beneficial to use language in this way, one *a priori* benefit of using language solely for exploration is that language is only needed during training, and not evaluation. Regardless, see Appendix E for additional experiments with MiniGrid agents that encode language into the state representation; while this boosts performance of baseline models, the experiments show that language-augmented exploration methods still outperform non-linguistic ones.

---

[5]For multiple messages, we average NovelD of each messsage.

[6]For an implementation of a message reward with simple novelty-based decay, see the message-only L-NovelD ablation results in Appendix F.2, which underperforms full L-NovelD and L-AMIGo.

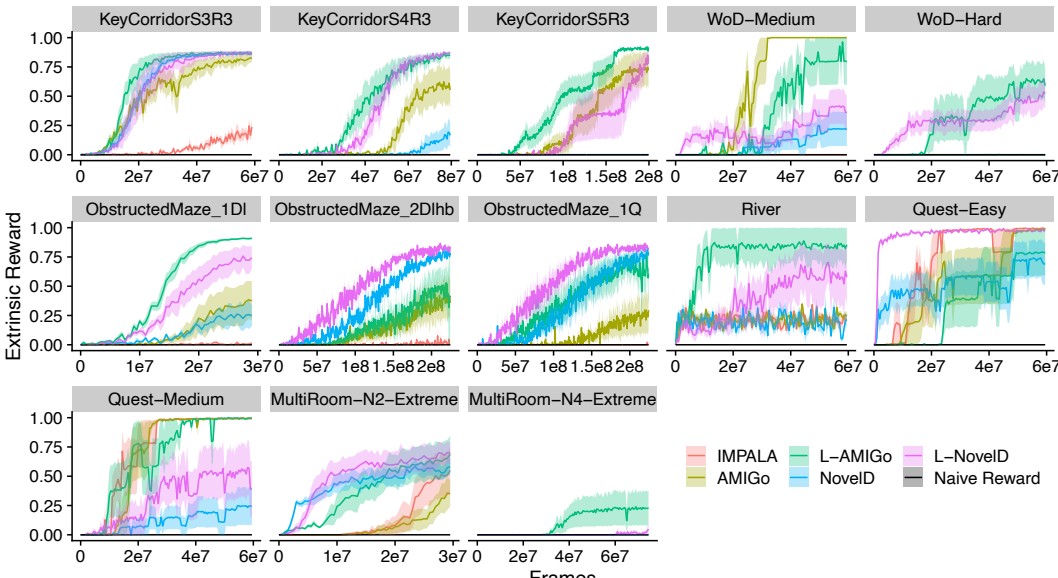

Figure 3: **Training curves**. Mean extrinsic reward ($\pm$ std err) across 5 independent runs for each model and environment. In general, linguistic variants outperform their non-linguistic forms.

**MiniHack.** MiniHack [41] is a suite of procedurally-generated tasks of varying difficulty set in the roguelike game NetHack [28]. MiniHack contains a diverse action space beyond simple MiniGrid-esque navigation, including planning, inventory management, tool use, and combat. These actions cannot be expressed by $(x, y)$ positions, but instead are captured by in-game messages (Figure 1). We evaluate our methods on a representative suite of tasks of varying difficulty: **River**, **Wand of Death (WoD)-{Medium,Hard}**, **Quest-{Easy,Medium}**, and **MultiRoom-{N2,N4}-Extreme**.

For space reasons, we describe the WoD-Hard environment here, but defer full descriptions of tasks (and messages) to Appendices G.2 and H. In WoD-Hard, depicted in Figure 1, the agent must learn to use a *Wand of Death*, which can zap and kill enemies. This involves a complex sequence of actions: the agent must find the wand, pick it up, choose to *zap* an item, select the wand in the inventory, and finally choose the zapping direction (towards the minotaur which is pursuing the player). It must then proceed past the minotaur to the goal to receive reward. Taking these actions out of order (e.g. trying to *zap* with nothing in the inventory, or selecting something other than the wand) has no effect.

It is difficult to enumerate all MiniHack messages, as they are hidden in low-level game code which has many edge cases. As an estimate, we can examine expert policies: agents which have solved WoD tasks encounter around 60 messages, of which only 5–10 (8–16%) are needed for successful trajectories, including inventory (*f - a metal wand*, *what do you want to zap?*, *in what direction?*) and combat (*You kill the minotaur!*, *Welcome to level 2.*) messages; most are irrelevant (e.g. picking up and throwing stones) or nonsensical (*There is nothing to pick up*, *That is a silly thing to zap*). In the other tasks, only 8–18% of the hundreds of unique messages are needed for success (Appendix G.2).

Unlike the MiniGrid environments, we adapt baseline models from [41], which all already encode the in-game message into the state representation. Despite this, as we will show, using language as an explicit target for exploration outperforms using language as a state feature alone.

## 7 Results

Figure 3 shows training curves with AMIGo, NovelD, language variants, and the IMPALA and naive message reward baselines. Following Agarwal et al. [2], we summarize these results with the interquartile mean (IQM) of all methods in Figure 4, with bootstrapped 95% confidence intervals constructed from 5k samples per model/env combination.[7][8] We come to the following conclusions:

---

[7]See Appendix D for full numeric tables and area under the curve (AUC)/probability of improvement plots.
[8]See Appendix F for ablation studies of L-AMIGo's grounding network and the components of L-NovelD.

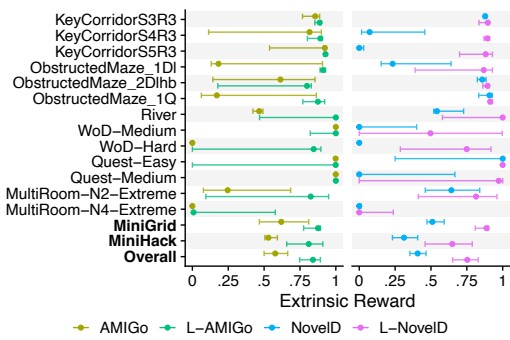
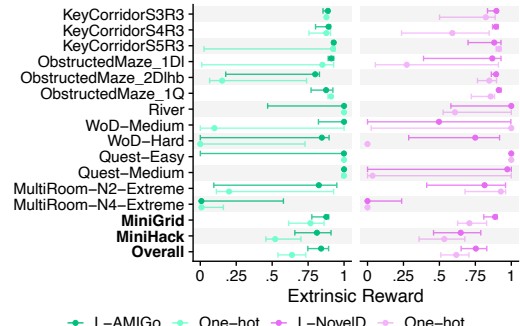

Figure 4: **Aggregate performance**. Interquartile mean (IQM) of models across tasks. Dot is median; error bars are 95% bootstrapped CIs.

Figure 5: **One-hot performance.** Models compared to variants with one-hot non-compositional goals. Plot elements same as Figure 4

**Linguistic exploration outperforms non-linguistic exploration.** Both algorithms, L-AMIGo and L-NovelD, outperform their nonlinguistic counterparts. Despite variance in runs and across environments, averaged across all environments (**Overall**) we see a statistically significant improvement of L-AMIGo over AMIGo (.27 absolute, 47% relative) and of L-NovelD over NovelD (.35 absolute, 85% relative). In some tasks, Figure 3 shows that L-AMIGo and L-NovelD reach the same asymptotic performance as their non-linguistic versions, but with better sample efficiency and stability (e.g. KeyCorridorS3R3 L-AMIGo, Quest-Easy L-NovelD; see Appendix D.3 AUC plots). Lastly, the failure of the naive message reward shows that indiscriminate reward shaping fails in tasks with sufficiently diverse language; instead, some notion of novelty or difficulty is needed to make progress.

**Linguistic exploration excels in larger environments.** Our tasks include sequences of environments with the same underlying dynamics, but larger state spaces and thus more challenging exploration problems. In general, larger environments result in bigger improvements of linguistic over non-linguistic exploration, since the space of messages remains relatively constant even as the state space grows. For example, there is no difference in ultimate performance for language/non-language variants on KeyCorridorS3R3, yet the gaps grow as the environment size grows to KeyCorridorS5R3, especially in L-NovelD. A similar trend can be seen in the WoD tasks, where AMIGo actually outperforms L-AMIGo in WoD-Medium, but in WoD-Hard is unable to learn at all.

### 7.1 Interpretability

One auxiliary benefit of our language-based methods is that the language states and goals can provide insight into an agents' training and exploration process. We demonstrate how L-AMIGo and L-NovelD agents can be interpreted in Figure 6.

**Emergent L-AMIGo Curricula.** Campero et al. [7] showed that AMIGo teachers produce an interpretable curriculum, with initially easy $(x, y)$ goals located next to the student's start location, and later harder goals referencing distant locations behind doors. In L-AMIGo, we can see a similar curriculum emerge through the proportion of *language goals* proposed by the teacher throughout training. In the KeyCorridorS4R3 environment (Figure 6a), the teacher first proposes the generic goal *open (any) door* before then proposing goals involving specific colored doors (where <C> is a color). Next, the agent discovers keys, and the teacher proposes *pick[ing] up the key* and putting it in certain locations. Finally, the teacher and student converge on the extrinsic goal *pick up the ball*.

Due to the complexity of the WoD-Hard environment, the curriculum for the teacher is more exploratory (Figure 6c). The teacher proposes useless goals at first, such as finding staircases and slings. At one point, the teacher proposes throwing stones at the minotaur (an ineffective strategy) before devoting more time towards wand actions (*you see here a wand*, *the wand glows and fades*). Eventually, as the student grows more competent, the teacher begins proposing goals that involve directly killing the minotaur (*you kill the minotaur*, *welcome to experience level 2*) before converging on the message *you see a minotaur corpse*—the final message needed to complete the episode.

**L-NovelD Message Novelty.** Similarly, L-NovelD allows for interpretation by examining the messages with highest intrinsic reward as training progresses. In KeyCorridorS4R3 (Figure 6b), the

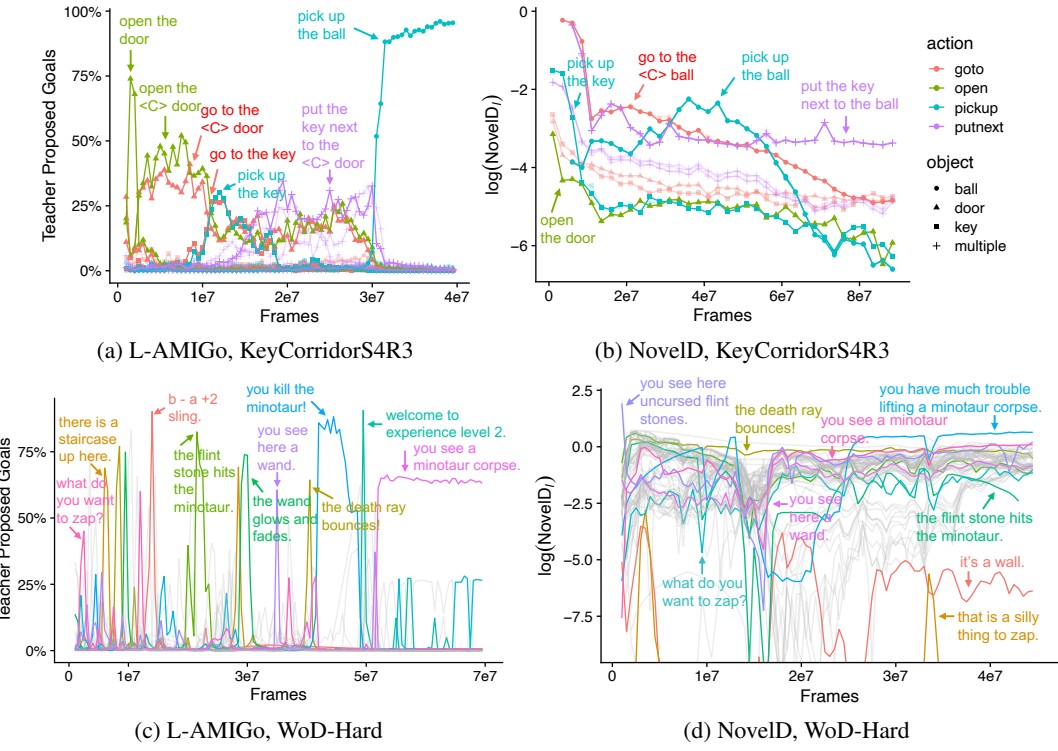

Figure 6: **Interpretation of language-guided exploration.** For the KeyCorridorS4R3 and WoD-Hard environments, shown are curricula of goals proposed by the L-AMIGo teacher (a,c) and the intrinsic reward of messages (some examples labeled) for L-NovelD (b,d).

novelty of easy goals such as *open the door* decreases fastest, while the novelty of the true extrinsic goal (*pick up the ball*) and even rarer actions (*put the key next to the ball*) remains high throughout training. In WoD-Hard (Figure 6d), messages vary widely in novelty: simple and nonsensical messages like *that is a silly thing to zap* and *it's a wall* quickly plummet, while more novel messages are rarer states that require killing the minotaur (*you have trouble lifting a minotaur corpse*).

## 7.2 Do semantics matter?

Language not only denotes meaningful features in the world; its lexical and compositional semantics also explain how actions and states relate to each other. For example, in L-AMIGo, an agent might more easily *go to the red door* if it already knows how to *go to the yellow door*. Similarly, in L-NovelD, training the RND network on the message *go to the yellow door* could lower novelty of similar messages like *go to the red door*, which might encourage exploration of semantically broader states. While our primary focus is not on whether agents can generalize to new language instructions or states, we are still interested in whether these semantics improve exploration for extrinsic rewards.

To check this hypothesis, in Figure 5 we run "one-hot" variants of L-AMIGo and L-NovelD where the semantics of the language annotations are hidden: each message is replaced with a one-hot identifier (e.g. *go to the red door* → 1, *go to the blue door* → 2) but otherwise functions identically to the original message. We make two observations. **(1)** One-hot goals actually perform quite competitively, demonstrating that the primary benefit of language in these tasks is to abstract over the state space, rather than provide fine-grained semantic relations between states. **(2)** Nevertheless, L-AMIGo is able to exploit semantics, with a significant improvement (.20 absolute, 32% relative) in aggregate performance over one-hot goals, in contrast to L-NovelD, which shows no significant difference. We leave for future work a more in-depth investigation into what kinds of environments and models might benefit more from language semantics.

# 8 Discussion

The key insight in this paper is that language, even if noisy and often unrelated to the goal, is a more abstract, efficient, and interpretable space for exploration than state representations. To support this, we have presented variants of two popular state-of-the-art exploration methods, L-AMIGo and L-NovelD, that outperform their non-linguistic counterparts by 47–85% across 13 language-annotated tasks in the challenging MiniGrid and MiniHack environment suites.

Despite their success here, our models have some limitations. First, as is common in work like ours, it will be important to alleviate the restriction on oracle language annotations, perhaps by using learned state description models [31, 49]. Furthermore, L-AMIGo specifically cannot handle tasks such as the full NetHack game which have unbounded language spaces and many redundant goals (e.g. *go to/approach/arrive at the door*), since it selects a single goal which must be achieved verbatim. An exciting extension to L-AMIGo would propose abstract goals (e.g. *kill [any] monster* or *find a new item*), possibly in a continuous semantic space, that can be satisfied by multiple messages.

More general extensions include better understanding when and why language semantics benefits exploration (Section 7.2) and using pretrained models to imbue semantics into the models beforehand [49]. Additionally, although the agents in this work are able to explore even when not all language is useful, we must take caution in adversarial settings where the language is completely unrelated to the extrinsic task (and thus useless) or even describes harmful behaviors. Future work should measure how robust these methods are to the noisiness and quality of the language. Nevertheless, the success of L-AMIGo and L-NovelD demonstrates the power of even noisy language in these domains, underscoring the importance of abstract and semantically-meaningful measures of exploration in RL.

## Acknowledgments and Disclosure of Funding

We thank Heinrich Küttler, Mikael Henaff, Andy Shih, Alex Tamkin, and anonymous reviewers for constructive comments and feedback, and Mikayel Samvelyan for help with MiniHack. JM is supported by an Open Philanthropy AI Fellowship.

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
