# OpenReview forum: "Improving Intrinsic Exploration with Language Abstractions"
_NeurIPS.cc/2022/Conference — NeurIPS 2022 Accept_

### Official Review · Reviewer_auiq · 2022-07-06

**Rating:** 6
**Confidence:** 4
**Soundness:** 3 good
**Presentation:** 3 good
**Contribution:** 2 fair

**Summary:**

The work proposes that the presence of (consistent) language, even if task-irrelevant and noisy, can provide a signal for exploration. They evaluate against two intrinsic exploration methods (Amigo and NovelD) by creating language abstraction variants of these methods. They run a thorough evaluation on mini-grid and minihack and find that exploration around language-based abstraction are indeed helpful.




**Questions:**

Interesting paper. Some points:

In your intro, you state that state-based exploration would reward going to every room but the agent should stay by the stove. Changing the objects on the stove achieves novel states so I'm not sure your example is correct. If we had access to ground-truth state, state-based exploration might very well lead to exploration by the stove. While I agree with your goal of using language, I don't see how what you said really supports language.

I could but wrong but it feels like you're using "state" in multiple ways: e.g., the observation the agent sees and underlying latent state that's generated the observation. Am I wrong about?

It seems a bit restrictive to also have an oracle grounding classifier. Do you ablate this?

Figure 5 has very similar colors. Overall, one-hot performance seems quite similar to using real-language. Does this change the story/message of your paper? The results are more-or-less the same indicating that it isn't necessarily language that's the driver of performance. Given that, I think your paper conclusions are not as well aligned with your story as you portray in the introduction. Don't get me wrong, I like this paper and think it's fairly thorough and clear but I think the findings don't align with the message. Could you clarify on this point?

The authors mention that there are several prior methods which use language for reward-shaping but they don't compare to any. How come?

Comment:
* Figure 6 has a lot going on. Challenging to make sense of.

Suggestions:
* Figure 2 is not as clear as it could be. Recommend describing it a bit more.
* Label "large" vs not large environments in Figure 3.


**Limitations:**

The limitations described seem adequate.

**Strengths And Weaknesses:**

Strengths:

* Related work is thorough and fairly clear.
* The methods are clear and clean extensions of prior work.
* The results are thorough and well-organized


Weaknesses:

* Related work is missing exploration based on state abstractions
* The contributions of this paper could be clarified

More on contribution:
The main finding is that exploration around language/state abstraction helps in large state spaces. We know that language abstraction helps in exploration--the authors reference many papers on this. In terms of state-abstraction helping in exploration, I think we also know this [1, 2]. The 1-hot id description conditioned student agent is like a parameterized skill that is being taught to go to abstract states. That said, I don't think state abstractions have been combined with these newer methods for exploration (Amigo/NovelD/etc.), so I think a discussion on this would clarify the contributions of this paper for me.



[1] Discovering Options for Exploration by Minimizing Cover Time
[2] Identifying useful subgoals in reinforcement learning by local graph partitioning

---

> ### Author Response · Authors · 2022-08-02
> **Response to reviewer auiq**
>
> Thanks to reviewer auiq for the thoughtful and helpful review!
>
> # On contributions
>
> One of reviewer auiq’s primary concerns is a clarification of the contributions of our paper. We recognize our contributions could be clarified, and will attempt to do so here (and in the paper):
>
> > The main finding is that exploration around language/state abstraction helps in large state spaces. We know that language abstraction helps in exploration--the authors reference many papers on this.
>
> We would like to clarify one point. As Section 2 states, while indeed there is substantial work in using language to guide exploration, it has nearly always been deployed in instruction following settings, where we have an oracle language instruction that can be used to shape rewards. Instead, we explore a new setting where we encounter many *intermediate* messages, associated with states, which have unknown usefulness for the extrinsic goal. This is a surprisingly realistic and understudied setting (please see our comments under **Dependence on an oracle language annotator** in response to Reviewer dbwv).
>
> **Thus, our paper makes a different, more specific claim about language and exploration**, which has not yet been made in the literature. Paraphrasing L324-325: “[intermediate, state-based] language, even if noisy and often unrelated to the goal…is [better for] exploration”. This claim, even if seemingly obvious in hindsight, is to our knowledge a novel one, which we have supported with algorithms and experiments in this paper. If reviewers know of existing work that tackles this setting, we invite them to clarify.
>
> > In terms of state-abstraction helping in exploration, I think we also know this [1, 2]. The 1-hot id description conditioned student agent is like a parameterized skill that is being taught to go to abstract states. That said, I don't think state abstractions have been combined with these newer methods for exploration (Amigo/NovelD/etc.), so I think a discussion on this would clarify the contributions of this paper for me.
>
> It is true that the 1-hot L-AMIGo student can be thought of as an AMIGo algorithm that simply proposes state abstractions as goals (without language). It is also true that, to our knowledge, no existing work shows how AMIGO/NovelD and other recent methods could be improved by incorporating state abstractions; however, as the reviewer correctly notes, it is well known that state abstractions can improve exploration.
>
> Our paper goes beyond this. We show that state abstractions can indeed improve the recent SoTA (AMIGo and NovelD), but crucially, that we also gain many additional benefits by using **language** as the medium for state abstraction. As we have discussed in the paper (and will discuss later in this response):
>
> - Language is a natural feature in many environments, and can be easily collected from humans and/or pretrained models.
> - Models, especially L-AMIGo in particular, can leverage the compositional semantics of the language to learn better.
> - Language improves the interpretability of the training process (Section 7.1).
>
> Reviewer auiq is correct that referencing the literature on state abstractions in RL will clarify our contributions and the precise role of language in the paper. We will add such references to the Related Work section of our paper (thank you to reviewer auiq for the references!). We hope this discussion has clarified the contributions of the paper and invite any further questions.
>
> # Other questions/comments
>
> > In your intro, you state that state-based exploration would reward going to every room but the agent should stay by the stove. Changing the objects on the stove achieves novel states so I'm not sure your example is correct. If we had access to ground-truth state, state-based exploration might very well lead to exploration by the stove. While I agree with your goal of using language, I don't see how what you said really supports language.
>
> While there may indeed be rewards for exploration near the stove, by looking at states alone, we argue there is no *a priori* reason to prefer (i.e. assign higher novelty to) exploration by the stove versus any other form of exploration, e.g. wandering off to the basement. Thus, we posit that the right kinds of language annotations for a domain could guide an agent towards desirable behaviors. This is similar to how the language descriptions in MiniHack (and indeed, human language) are biased towards describing meaningful actions like inventory manipulation and combat, rather than making random sequences of “rotate” actions or endlessly bumping into walls.
>
> Of course, there could be adversarial settings where the language is completely unrelated to the extrinsic goal, as we discuss in L339, which we do not explore. Our base assumption is that the language, while noisy, does at least partially include behaviors relevant for the extrinsic goal.

---

> > ### Author Response · Authors · 2022-08-02
> > **Response to reviewer auiq (continued)**
> >
> > > I could but wrong but it feels like you're using "state" in multiple ways: e.g., the observation the agent sees and underlying latent state that's generated the observation. Am I wrong about?
> >
> > As we assume a typical (fully observed) MDP, our use of the word “state” assumes that the observation that the agent “sees” and the “state” of the environment/game, are the same. We make no distinction between “states” and “(partial) observations”, like in a POMDP. There may indeed be a latent game state (e.g. the latent representation of the world in the MiniHack game engine), but we do not consider this in our framework. We will go through and make sure our wording is more precise.
> >
> > > It seems a bit restrictive to also have an oracle grounding classifier. Do you ablate this?
> >
> > **We do not use an oracle grounding classifier in any of our experiments.** Indeed, learning this classifier is one of the primary challenges we faced in transitioning from AMIGo to L-AMIGo, that we believe is helpful for the research community to know! In L167 we say that “Given an oracle grounding classifier…this is equivalent…**In practice, however, we learn the classifier online**”. L169-175 and Figure 2ii explain how the classifier is trained online from messages dynamically collected by the student. For ablations where we remove the grounding network entirely, please see Appendix E.1. We recognize the ambiguity here and will clarify this in the paper.
> >
> > > Figure 5 has very similar colors. Overall, one-hot performance seems quite similar to using real-language. Does this change the story/message of your paper? The results are more-or-less the same indicating that it isn't necessarily language that's the driver of performance. Given that, I think your paper conclusions are not as well aligned with your story as you portray in the introduction. Don't get me wrong, I like this paper and think it's fairly thorough and clear but I think the findings don't align with the message. Could you clarify on this point?
> >
> > This is a good question. First, **we disagree that the results are “more or less the same.”** L-AMIGo improves over one-hot L-AMIGo by **32%** (.20 absolute), and this difference is statistically significant. L-NovelD also seems to improve slightly over one-hot goals by **22%** (.14 absolute), but this difference is not statistically significant given the variance in our experiments. Thus, one conclusion we draw is that L-AMIGo is better than L-NovelD at leveraging the compositionality of language to improve exploration (L319).
> >
> > Regardless, we do agree that performance with one-hot language descriptions is quite competitive. This indicates that language provides two benefits:
> > - Coarsens/abstracts the state space; i.e. the presence of language highlights when states are relevant, regardless of what the language actually “means”. This is the same benefit that more general (one-hot) state abstractions give, as we have discussed above.
> > - Describes fine-grained and compositional semantic relations between states. In other words, if two states have similar/different language descriptions, then they are similar/different, and exploration methods might be able to take advantage of this.
> >
> > Our results show that indeed, language benefits via (1) coarsening the state space, but that L-AMIGo in particular is also able to leverage language compositionality (2) to learn more effectively.
> >
> > **Note that even if language is only beneficial to coarsen the state space, there is still value in using language over one-hot or other features, for the following reasons:**
> >
> > - First, language is common; see our response to reviewer dbwv on environments that have oracle language annotators.
> > - Second, it is easy to collect from human annotators: it is easy to ask someone to simply “describe a scene” in unrestricted language, rather than provide a specialized interface for annotating states.
> > - Third, we have an abundance of multimodal pretrained models that can generate language description of states automatically (e.g. as used in [Zeng et al., 2022](https://arxiv.org/abs/2204.00598)), and thus developing methods that can handle such language annotations is especially important.
> > - Finally, language improves interpretability of the training process, as discussed in Section 7.1.

---

> > > ### Author Response · Authors · 2022-08-02
> > > **Response to reviewer auiq (continued)**
> > >
> > > > The authors mention that there are several prior methods which use language for reward-shaping but they don't compare to any. How come?
> > >
> > > **This is incorrect;** we do compare to existing naive methods for reward shaping in Figure 3 (black line; Naive Reward). More generally, however, this is a moot comparison, because there are no existing methods for language exploration applicable in our problem setting, and (as we have shown) trying to adapt existing methods fails.
> > >
> > > We have described why in Section 2 and Appendix B, but to reiterate:
> > > existing language reward-shaping methods are nearly always designed for instruction following settings, where a language description of the extrinsic goal can be used to shape rewards towards that goal. In our setting, however, we do not have an extrinsic goal, but rather many *intermediate* messages, associated with states, that have unknown usefulness for the extrinsic goal.
> > >
> > > The problem with the existing work is that because they assume a single high-level instruction, **they assume language is always helpful.** For example, LEARN ([Goyal et al., 2019](https://www.ijcai.org/proceedings/2019/0331.pdf)) proposes to assign rewards proportional to the likelihood that a state is relevant to the extrinsic (language) goal. In our setting, however, we have no extrinsic goal, and instead directly encounter states associated with language. Naively applying existing methods like LEARN to our setting, therefore, would mean simply rewarding an agent for every message it encounters in the environment. This is precisely the Naive Reward baseline in Figure 3, which completely fails to learn. Even adding novelty-based decay to this naive message reward (the language-only L-NovelD baseline in Appendix E.2) underperforms.
> > >
> > > The key innovation that our algorithms have, then, is learning *when* to reward certain language states over others, either via an adversarial teaching process (L-AMIGo) or by gradually decaying reward as training progress (L-NovelD). As our experiments in Figure 3 show, this is crucial for obtaining non-trivial performance on our tasks.
> > >
> > > Again, we have more discussion in Section 2 and Appendix B, have described the existing work that we are aware of, and explained how they are not applicable in our problem setting. If reviewer auiq or others think we have missed some relevant work, we invite them to clarify.
> > >
> > > # Summary
> > >
> > > We hope our response addresses your concerns. Please let us know if you have any further questions. Given this additional discussion, especially on the clarification of contributions of our paper, we hope you consider raising your support for our paper.

---

> > > > ### Comment · Reviewer_auiq · 2022-08-08
> > > > **Thanks for the response**
> > > >
> > > > Thank you for the response.
> > > >
> > > > Point taken that a contribution of this paper is that language as a state-abstraction helps for exploration. I think this is expected behavior, but this is still evidence in that direction.
> > > >
> > > >
> > > > Regarding whether L-Amigo can leverage the compositional semantics of language better, I suppose the results ablating to a 1-hot encoding and getting a 32% improvement is evidence for this? I agree that this points in that direction. I think the paper would benefit from an explanation of why L-Amigo can better leverage compositional semantics of language. Given what you said, it seems that L-NovelID is benefitting more-so from state-abstractions than from language since it isn't a statistically significant improvements in performance. I agree with your characterization: "language benefits via (1) coarsening the state space, but that L-AMIGo in particular is also able to leverage language compositionally (2) to learn more effectively." I hope you add that to the main paper.
> > > >
> > > >
> > > > Point well-taken that language is common, easy to collect, and improves interpretability. 1-hot encodings are as well though, just ask someone to increase a counter each time a new 'state' is encountered.
> > > >
> > > > Overall, while I don't change my score, I still recommend acceptance.

---

### Official Review · Reviewer_c6cu · 2022-07-09

**Rating:** 7
**Confidence:** 4
**Soundness:** 3 good
**Presentation:** 3 good
**Contribution:** 2 fair

**Summary:**

This paper proposes the use of _language_ to improve exploration via intrinsic rewards. The paper proposes two methods that make use of sentences describing the goals in particular states.

**Questions:**

- Do the baselines make any use of the descriptive sentences in the environment? Is it part of the environment state? As mentioned above, I am concerned that the increased performance stems from the high quality of the language annotations and not from the methodology itself. Could the authors elaborate on this?
- In Discussion (Sec 8), the authors mention that it might be possible to lift the restriction of an oracle language annotator by using a learned state description. Instinctively, I would say that since this is used as an intrinsic reward, such a model would require multiple visitations to that particular state anyway and would lose its novelty along the way. Am I misunderstanding something? If yes, such a result would be truly impressive and an important contribution.


**Limitations:**

Yes, the authors have addressed the limitations and potential negative societal impact of their work.


**Strengths And Weaknesses:**


Strengths:

- The paper is very well written, and clearly motivates the problem and solution.
- Exploration in RL is an ever-important problem, and the paper proposes a solution that improves upon state-of-the-art methods (but with some limitations, like the assumption of an "oracle language annotator")
- An impressive set of results in two significantly hard environments (however, I have some concerns about the experiments)

Weaknesses:

- L-AMIGo and particularly L-NovelD seem to be significantly similar to their respective base algorithms, and just offer a way to use state information in the form of language.
- L-NovelD uses NovelD(s_t, s_{t-1}) (eq7) which I find slightly concerning. I would hope that NovelD(l_t, l_{t-1}) would give a strong learning signal by itself, but as the authors explain it does not lead to equally high performance. This ties to my main concern (below) that the two methods mostly benefit from the (extra) data provided by the oracle and not because they are by themselves enough to provide an intrinsic reward.
- My main concern is that the baselines (AMIGo/NovelD/IMPALA) do not use the information used by their L- extensions. After reading the paper, it is still not clear to me what the main contributing factor to the improved performance is (also see questions below).

I have enjoyed reading this paper, but I believe it needs to address the issue of my last bullet point. Even a comparison to baselines that at least make use of the generated sentences in a naive way would be somewhat useful.

I would be open to increasing my score if the authors give a satisfying answer to my concerns.

** Update after rebuttal: This reviewer has increased the overall rating from 4 to 7. **

---

> ### Author Response · Authors · 2022-08-02
> **Response to reviewer c6cu**
>
> Thank you to reviewer c6cu for the insightful and helpful review!
>
> # On baselines using language features
>
> Reviewer c6cu’s main concern is a very sensible one:
>
> > My main concern is that the baselines (AMIGo/NovelD/IMPALA) do not use the information used by their L- extensions. After reading the paper, it is still not clear to me what the main contributing factor to the improved performance is (also see questions below).
>
> > Do the baselines make any use of the descriptive sentences in the environment? Is it part of the environment state? As mentioned above, I am concerned that the increased performance stems from the high quality of the language annotations and not from the methodology itself. Could the authors elaborate on this?
>
> Thanks to reviewer c6cu for this very astute observation! Our response:
>
> ## MiniHack
>
> **In MiniHack, all methods and baselines encode the language annotations into their policy!** We apologize for being unclear about this. As mentioned in L226, we extend [Samvelyan et al. (2021)’s](https://arxiv.org/abs/2109.13202) baseline model which indeed encodes the language string as input. **Thus, the MiniHack results show it is not sufficient to simply use the message as a feature. Instead, the full benefits of language come when it is used as an explicit target for intrinsic exploration.**
>
> ## MiniGrid
>
> In MiniGrid, the baselines reported in the main results of the paper (Figure 3, Figure 4) do *not* encode the language annotations into their policy. Thus, we reran 5 seeds for each of the MiniGrid models (IMPALA, AMIGo, L-AMIGo, NovelD, L-NovelD), where the language representation is encoded into the state.
>
> Note that running NovelD and L-NovelD with the language representation encoded into the state is very similar to the experiments we have already run in Appendix E.2, where we compared L-NovelD to a version of NovelD where the language is combined with the state representation. L-NovelD is somewhat redundant in this setting, since it defines a novelty term both on the combined (state, language) representation and the language representation alone. Regardless, in Appendix E.2, L761-766 we noted that L-NovelD and NovelD (combined) reached similar performance for MiniGrid tasks, but not MiniHack. We wrote:
>
> > …while combining the state and language into a single embedding works well for MiniGrid, it does not work as well for MiniHack tasks, suggesting that the additional flexibility afforded by the separate L-NovelD term can be helpful in many settings…*the point of L-NovelD is to clarify the contributions made by both the language and the state and make it easier to trade-off between the two.*
>
> The only difference between the models in Appendix E.2. and the new NovelD experiments we present here is that the combined (state, language) representation is not just used as a novelty signal, but also used by the policy and value heads. However, given these results, we would expect (and will show) that the results are roughly the same.
>
> **Please see the file we have added to the supplementary material (REBUTTAL.pdf) for a before/after comparison of Figures 3 and 4, where we have updated both figures with the new baselines. We also include the raw numbers here, plus 95% confidence intervals, for convenience:**

---

> > ### Author Response · Authors · 2022-08-02
> > **Response to reviewer c6cu (continued)**
> >
> > ## Before
> >
> > | Environment           | AMIGo             | L-AMIGo                    | NovelD                     | L-NovelD                   |
> > |-----------------------|-------------------|----------------------------|----------------------------|----------------------------|
> > | KeyCorridorS3R3       | 0.86 (0.77, 0.89) | 0.89 (0.85, 0.90)          | 0.88 (0.87, 0.88)          | **0.90** (0.83, 0.90) |
> > | KeyCorridorS4R3       | 0.82 (0.11, 0.90) | **0.89** (0.80, 0.91) | 0.07 (0.02, 0.45)          | **0.89** (0.87, 0.91) |
> > | KeyCorridorS5R3       | 0.92 (0.54, 0.93) | **0.93** (0.92, 0.93) | 0.00 (0.00, 0.03)          | 0.88 (0.70, 0.93)          |
> > | ObstructedMaze\_1Dl   | 0.18 (0.13, 0.91) | **0.91** (0.89, 0.93) | 0.23 (0.15, 0.64)          | 0.87 (0.39, 0.93)          |
> > | ObstructedMaze\_2Dlhb | 0.61 (0.14, 0.86) | 0.80 (0.18, 0.83)          | 0.86 (0.82, 0.88)          | **0.89** (0.86, 0.90) |
> > | ObstructedMaze\_1Q    | 0.17 (0.06, 0.86) | 0.88 (0.77, 0.92)          | **0.91** (0.83, 0.93) | **0.91** (0.91, 0.93) |
> > | **MiniGrid**     | 0.62 (0.47, 0.81) | 0.88 (0.78, 0.89)          | 0.51 (0.47, 0.59)          | **0.89** (0.81, 0.90) |
> > | **Overall** (i.e. w/ MiniHack)      | 0.57 (0.50, 0.66) | **0.84** (0.74, 0.89) | 0.41 (0.35, 0.47)          | 0.76 (0.66, 0.83)          |
> >
> > ## After
> >
> > | Environment             | AMIGo             | L-AMIGo           | NovelD            | L-NovelD          |
> > |-----------------------|-------------------|-------------------|-------------------|-------------------|
> > | KeyCorridorS3R3       | 0.88 (0.88, 0.89) | 0.88 (0.88, 0.89) | **0.89** (0.88, 0.89) | **0.89** (0.88, 0.89) |
> > | KeyCorridorS4R3       | 0.88 (0.79, 0.90) | 0.89 (0.88, 0.89) | 0.89 (0.85, 0.90) | **0.90** (0.89, 0.91) |
> > | KeyCorridorS5R3       | 0.92 (0.90, 0.93) | **0.93** (0.92, 0.93) | 0.01 (0.00, 0.07) | 0.89 (0.07, 0.93) |
> > | ObstructedMaze_1Dl   | 0.91 (0.41, 0.93) | 0.92 (0.90, 0.93) | **0.93** (0.91, 0.93) | **0.93** (0.92, 0.93) |
> > | ObstructedMaze_2Dlhb | 0.81 (0.34, 0.87) | 0.86 (0.83, 0.87) | **0.89** (0.85, 0.91) | 0.87 (0.74, 0.88) |
> > | ObstructedMaze_1Q    | 0.92 (0.90, 0.93) | 0.91 (0.88, 0.92) | 0.92 (0.91, 0.94) | **0.93** (0.93, 0.94) |
> > | **MiniGrid**              | 0.88 (0.79, 0.90) | **0.90** (0.89, 0.90) | 0.75 (0.74, 0.76) | **0.90** (0.76, 0.91) |
> > | **Overall** (i.e. w/ MiniHack)              | 0.69 (0.65, 0.73) | **0.85** (0.77, 0.90) | 0.52 (0.47, 0.57) | 0.76 (0.65, 0.83) |

---

> > > ### Author Response · Authors · 2022-08-02
> > > **Response (continued)**
> > >
> > > **Our updated conclusions for MiniGrid are:**
> > >
> > > - **Incorporating language into the feature space improves performance.** Indeed, as reviewer c6cu notes, the quality of the language annotations in MiniGrid are helpful for learning. All models’ performance has increased as a result of the linguistic annotations.
> > > - **However, just incorporating language into the feature space is insufficient for competitive performance.** While the IMPALA baseline is able to make more progress in the simpler environments (e.g. now nearly solving KeyCorridorS3R3), it still lags behind and is unable to solve the other environments. **This clearly illustrates that in order to reap the benefits of language, it is important to use it in conjunction with exploration, not just as a feature.**
> > > - **L-AMIGo continues to outperform AMIGo, though the differences are smaller.** While the tables show that both L-AMIGo and AMIGo can solve each task, reaching roughly similar final performance, they differ in sample efficiency as well as training stability (examine Figure 3 in REBUTTAL.pdf, and error bars in the tables and Figure 4). Similar to what we noted in the paper (L268-269), the full training curves in the updated Figure 3 show that while L-AMIGo and AMIGo reach similar asymptotic performance, L-AMIGo learns quicker or more stably. This happens quite clearly in KeyCorridorS5R3 and the Maze environments.
> > > - **L-NovelD and NovelD performance on MiniGrid is closer when language is combined with the state representation,** except on a few tasks: e.g. KeyCorridorS4R3, and on KeyCorridorS5R3 where NovelD is still unable to learn.
> > >
> > >   Again, these NovelD results are somewhat unsurprising given what we have already said in Appendix E.2. As reviewer c6cu notes, L-NovelD is not fundamentally different from NovelD; it is simply an adaptation of NovelD that lets us precisely tradeoff between language and state novelty. By comparing L-NovelD (with language in the state representation) to NovelD (with language in the state representation), we are comparing essentially the same model.
> > >
> > >   Our goal is not for L-NovelD to “win” over NovelD; if we were able to consistently benefit from language by simply incorporating it into the state (which works for MiniGrid), we would have simply proposed to do that; however, our MiniHack experiments in Figure 3 and Appendix E.2 show that this is not always the case, and it is helpful to have the “additional flexibility afforded by the separate L-NovelD term” (L763).
> > >
> > >   **To summarize:** these new results, where L-NovelD and combined NovelD outperform IMPALA, show that language is important for exploration; the MiniHack results, especially, show the same thing, *but additionally that L-NovelD can be preferable to NovelD.*
> > >
> > > Note that due to time and compute constraints we were not able to recreate every figure in the paper and appendix with these new models, but will do so (and incorporate our new results into Figures 3 and 4) for the camera-ready version, if accepted.

---

> > > > ### Author Response · Authors · 2022-08-02
> > > > **Response (continued)**
> > > >
> > > > ## Conclusion (baselines)
> > > >
> > > > Again, thanks to c6cu for noticing this, and prompting us to run additional experiments, which we believe have strengthened the baselines, and thus quality, of the paper.
> > > >
> > > > **Again, keep in mind the majority of our experiments remain unchanged.** We will incorporate the new results into our paper: our MiniGrid results have become admittedly subtler, while our MiniHack results remain the same. **Overall, the conclusions of our paper are unchanged:** language outperforms states as a tool for exploration. L-AMIGo outperforms AMIGo by **23%** (.16 absolute), and L-NovelD outperforms NovelD by **46%** (.24 absolute), across the environments we tested. **Both differences remain statistically significant,** as the Tables above, and Figure 4 in REBUTTAL.pdf, shows. **We have updated our abstract on OpenReview,** and will go through and update the rest of the paper for the camera ready revision.
> > > >
> > > > Moreover, given these new experiments, we make an additional claim: **while incorporating language into the state can give improvements, the full benefits of language come when it is used not just as an additional feature, but as an explicit target for exploration.**
> > > >
> > > > However, we will point out that our new results are less dramatic for MiniGrid than MiniHack, now that all models encode language. We’ll include a discussion of why this is: in general, we believe MiniHack is a much harder environment suite, with a richer state space, where it is more important for language to abstract over the space. Moreover, the differences between L-NovelD and NovelD become especially important when moving from MiniGrid to MiniHack.
> > > >
> > > > We hope this has alleviated reviewer c6cu’s concerns and hope they increase their support for the paper, or otherwise elaborate what concerns remain.
> > > >
> > > > # Other questions/comments
> > > >
> > > > > L-AMIGo and particularly L-NovelD seem to be significantly similar to their respective base algorithms, and just offer a way to use state information in the form of language.
> > > >
> > > > Please see our response on Novelty we made to Reviewer dbwv. To summarize: our hypothesis is that language serves as an alternative target for exploration than states. To evaluate this, we should extend existing methods, rather than invent new methods that are completely ungrounded from the extensive literature on state-based exploration. In fact, as reviewer auiq notes, it is a **strength** that our methods are “clean and clear extensions of prior work.” Finally, such extensions are non-trivial, and we believe the experimental results and ablations we provide are valuable to the community.
> > > >
> > > > > L-NovelD uses NovelD(s_t, s_{t-1}) (eq7) which I find slightly concerning. I would hope that NovelD(l_t, l_{t-1}) would give a strong learning signal by itself, but as the authors explain it does not lead to equally high performance.
> > > >
> > > > This is correct, though we believe it is a feature, not a bug, that language- and state-based exploration can be combined for orthogonal benefits. Keep in mind that we present not just L-NovelD, but also L-AMIGo, and L-AMIGo does not use (x, y) goals or any supplemental state-based novelty measure (though it is certainly possible that a supplemental measure would improve L-AMIGo further).
> > > >
> > > > Thus, one interpretation of these results is that L-AMIGo is able to more effectively explore with language alone than L-NovelD, because L-AMIGo is more selective in choosing *which* language descriptions to reward. In contrast, L-NovelD indiscriminately rewards all messages, with only a decaying novelty signal to encourage the agent to find other messages.
> > > >
> > > > > In Discussion (Sec 8), the authors mention that it might be possible to lift the restriction of an oracle language annotator by using a learned state description. Instinctively, I would say that since this is used as an intrinsic reward, such a model would require multiple visitations to that particular state anyway and would lose its novelty along the way. Am I misunderstanding something? If yes, such a result would be truly impressive and an important contribution.
> > > >
> > > > We are not sure we understand Reviewer c6cu’s question here. Our idea was to use a state description model trained offline on (state, language description) pairs, which is then frozen and deployed during training. Thus the algorithm(s) would be identical to what we have presented, except instead of language descriptions given from the environment, they are generated from a learned model instead (see our response to reviewer dbwv under **Dependence on an oracle language annotator** for concrete ways to do this). We invite Reviewer c6cu to clarify this question.
> > > >
> > > > # Summary
> > > >
> > > > Again, we thank you for (rightfully) bringing up concerns about the baselines used in this work, and we hope our response and additional experiments have addressed your concerns. Please let us know if you have any further questions. Given the additional experiments and discussion, we hope you consider raising your support for our paper.

---

> > > > > ### Comment · Reviewer_c6cu · 2022-08-03
> > > > > **A clarification of my comment**
> > > > >
> > > > > > We are not sure we understand Reviewer c6cu’s question here
> > > > >
> > > > > Apologies for not being sufficiently clear and thank you for explaining further.
> > > > > What I meant is that collecting "(state, language description) pairs" to train offline is not trivial at all. Collecting the state data requires exploration of the environment (the problem we are trying to solve with intrinsic exploration in the first place). In any case, I understand this is future work and not related to the paper's contribution. I will not base my updated evaluation of this paper on this discussion.

---

> > > > > > ### Author Response · Authors · 2022-08-07
> > > > > > **Response**
> > > > > >
> > > > > > Yes, this is a great point! If we had access to unlabeled states in our task of interest, we imagine it would be straightforward to ask for human/model language descriptions. However, if we do not even have trajectories or states to annotate, we completely agree that jointly collecting state and language data is a non-trivial "chicken-and-egg" problem. One would hope that we can use models to generate language descriptions on-the-fly, but this is limited to settings where we have such a grounded language model (e.g., a photorealistic environment where we could transfer from pretrained models), but we agree this is an important avenue for future work.
> > > > > >
> > > > > > Since the discussion period is coming to a close, we would like to know if our updated experiments and discussion have changed your evaluation of our paper. Again, thank you for suggesting the additional experiments, which we believe have improved the quality and rigor of our work. As we mentioned, we are happy to continue discussing and refining our conclusions, if you have any remaining concerns.

---

> > > ### Comment · Reviewer_c6cu · 2022-08-03
> > > **Minor follow up question on table above.**
> > >
> > >
> > > Could the authors clarify the use of *bold* in the above table? Are those statistically significant or simply the highest values? To me, it looks like differences in MiniGrid are not statistically significant (with the exception of S5R3 in NovelD/L-NovelD).
> > >
> > > If so, and from the conclusion below,
> > > > Overall, the conclusions of our paper are unchanged: language outperforms states as a tool for exploration. L-AMIGo outperforms AMIGo by 23% (.16 absolute), and L-NovelD outperforms NovelD by 46% (.24 absolute), across the environments we tested. Both differences remain statistically significant...
> > >
> > > Should this be rephrased to say that these significant differences were only found in the MiniHack environment and not Minigrid (esp. in AMIGO/L-AMIGO)? I think it is important for the reader to understand if your conclusions apply generally or whether they only apply to a specific environment (MiniHack).

---

> > > > ### Author Response · Authors · 2022-08-03
> > > > **Response**
> > > >
> > > > Thanks for this question! Our use of bold above simply refers to the highest performing model in each row, without regards to statistical significance. Reviewer is correct that many of the differences in performance in models here are not statistically significant, at least partially because all models are ultimately reaching roughly ceiling performance on the task.
> > > >
> > > > However, as we discussed in the response, **these performance numbers do not tell the whole story**: there is a clear difference in performance when one looks not just at the final performance numbers, but the actual learning curves (and stability of these curves). Again, please see Figure 3 which shows that L-AMIGo reaches the same performance as AMIGo, but quicker in some environments (KeyCorridorS5R3, Maze) environments, and Figure 4 which demonstrates the larger variance present in AMIGo results.
> > > >
> > > > We believe that due to the increased stability and sample efficiency of AMIGo as apparent from Figures 3 and 4, we can still say that L-AMIGo appears to do better in MiniGrid, but we will note in the main text that these results are admittedly subtler, and the difference is not necessarily in final performance, but in sample efficiency/stability.
> > > >
> > > > Please let us know what you think of this conclusion in light of what we have said above - if you still disagree with the way we are phrasing the conclusion, we are happy to update our writing.

---

> ### Comment · Reviewer_c6cu · 2022-08-07
> **Update after author rebuttal**
>
> I would like to thank the authors for their responses. The authors addressed most of my concerns. Specifically:
>
> > "L-" algorithm extensions using more data.
>
> This was not the case for "MiniHack", but it was for "MiniGrid". However, the authors run new experiments on the "MiniGrid" environment that made the comparison fairer ensuring the language representation is encoded into the state. I hope the authors will also include a more detailed explanation of how the representation is encoded in the main text (or appendix) as they see fit and their updated conclusions (seen in their response below) for MiniGrid in the main text.
>
> These completely address my original concern and are significant improvements over the original presentation.
>
> > L-NovelD uses NovelD(s_t, s_{t-1}) (eq7)
>
> > L-AMIGo and particularly L-NovelD seem to be significantly similar to their respective base algorithms.
>
> After the author's explanations, I better understand their motivations and I agree that the above points are not particularly important. This motivation was explained with the answer below: "our hypothesis is that language serves as an alternative target for exploration than states". When re-visiting the paper I believe this was explained, but I understand how it can be missed by a reader that focuses on the experimental results and improvements of the underlying algorithms. In my opinion this is a clear problem a motivation and the experimental results and analysis are of high quality and can be helpful to the community.
>
> A concern that might arise after the new set of results is that improvements in the MiniGrid are not particularly significant.
>
> However, there are some notable exceptions: In "KeyCorridorS5R3", language annotations make the difference between the environment being solved or not. In "Amigo/L-Amigo" there are differences in the stability of the runs which signifies a difference caused by the respective algorithms. The authors give reasonable analysis and explanations for these numbers and I am quite satisfied by their arguments. I would urge the authors to differentiate the MiniHack and MiniGrid environments when discussing statistically significant results in the text (if any such discussions remain).
>
> Perhaps it would be interesting for the authors to explore slightly harder MiniGrid environments (if any exist) in case they find significant differences there.
>
> Finally, I will have to note that I personally do not weight in my final score state-of-the-art results as much as a novel, clear, and insightful analysis of the hypothesis and results. Please, see my original review for an updated score.

---

### Official Review · Reviewer_dbwv · 2022-07-11

**Rating:** 4
**Confidence:** 4
**Soundness:** 3 good
**Presentation:** 3 good
**Contribution:** 2 fair

**Summary:**

This paper posits the hypothesis that language abstractions are superior to raw states to guide exploration motivated by intrinsic reward.

The authors extend two recent and performant algorithms, AMIgo and Nove1D, to use language abstractions instead of states as a means of computing intrinsic rewards.

They show on a suite of experiments that, indeed, these language-based algorithms outperform their state-based counter-parts thereby supporting their hypothesis.

Moreover, while previous works using language as an abstraction for exploration have focused only on comparing their methods to vanilla RL, here a useful systematic study of the performance of language abstraction vs. state based intrinsic reward exploration is presented.



**Questions:**

1. In L-AMIgo, I’m curious about the impact of the “grounding network”. Is it critical performance? Did you by any chance do ablation study on the components?

**Limitations:**

In Section 8, the authors are up-front about the fact that the L-variants described in their paper is so far only applicable in environments that come with a language "oracle". Indeed, this seems like a significant limitation that perhaps explains why most literature focuses on state based measure for generating intrinsic rewards.

**Strengths And Weaknesses:**

**Strengths**

1. In terms of quality and soundness, I cannot find much to fault with the paper. The claims of the paper are clearly stated and quite strongly supported by empirical results (v. Figure 3). That is, indeed language abstractions appear to be better than state as a basis for generating intrinsic reward.
2. In terms of clarity,  the paper is largely clear and concise. It is easy to follow the development of the logic throughout. The authors carefully provide all necessary details to understand the development of their new algorithms. Moreover, all figures, captions, and all plots in general are clean and easy to understand.

**Weaknesses**

1. In my opinion, the originality of this paper is limited. It could be argued that replacing state as the means of computing intrinsic reward with language amounts to simply changing the input space of AMIgo and Nove1D. I do recognize that some innovations are required for the adaptation of the base algorithms to language abstractions (e.g., the grounding network), but I am not convinced that these are large novel steps.
2. Notwithstanding the above, if there was a method to develop language abstractions for environments in general, the applicability of the algorithms developed in this paper may be rationalized. However, the L-variants are limited to environments which come pre-packaged with language abstractions thereby limiting their scope. In fact, in Zhang et al. [1], Nove1D is flexibly used in environments which have language abstractions (MiniGrid) and which don’t have language abstractions (Atari), while L-Nove1D cannot be used as flexibly.

Overall, my main contention is that the originality and novelty are somewhat limited.

[1] T. Zhang, H. Xu, X. Wang, Y. Wu, K. Keutzer, J. E. Gonzalez, and Y. Tian. NovelD: A simple yet effective 478 exploration criterion. In Advances in Neural Information Processing Systems, 2021.

---

> ### Author Response · Authors · 2022-08-02
> **Response to reviewer dbwv**
>
> Thank you to reviewer dbwv for the insightful review, and for engaging with our paper!
>
> # On Novelty
>
> **Note that reviewer dbwv’s main issue with the paper is simply one of limited novelty.** Otherwise, reviewer dbwv finds the paper “clear and concise”, “strongly supported by empirical results”, and overall “cannot find much to fault with the paper”.
>
> We view our paper as making three contributions:
>
> - **Conceptual.** A primary contribution of this paper is the insight that language, even if noisy and task-agnostic, serves as a better (or additional) space for exploration than nonlinguistic state representations (L324-325). This is a **novel** idea, with a **novel problem setting** that has not existed in the literature. As we describe in Sec 2 related work, existing approaches to language-guided exploration are designed for instruction following, where there is a single oracle language instruction. Instead, we introduce a new setting where we encounter many *intermediate* language commentary, associated with states, that have unknown relation to the true goal. This is a surprisingly common, yet understudied, scenario (see our comments on **Dependence on an oracle language annotator** below). We explain how existing methods cannot be applied in Sec 2 and show that naive versions of existing methods fail in Figure 3 and Appendices B and E.2.
> - **Algorithmic.** To evaluate the conceptual hypothesis above, we present not one, but *two* algorithms extending the existing state-of-the-art in state-based exploration to language abstractions: L-AMIGo and L-NovelD. Since our hypothesis is that language can serve as an alternative space for exploration than states, *we deliberately build on and compare to prior work*; more on this later.
> - **Empirical.** Conceptual and algorithmic contributions alone are not enough to prove the idea; they need to be supported by empirical results. To do so, we provide extensive experiments and ablations comparing exploration methods with and without language, across 13 tasks in 2 challenging domains. (Note that on the empirical side, reviewer dbwv finds the contribution “strong”.)
>
> It seems reviewer dbwv’s primary concern is the novelty of the **algorithmic** contribution of our work: that it builds on existing work and has limited novelty beyond the existing work.
>
> **This is intentional.** The core insight of our paper is that when used for exploration, **language should be treated the same as state representations.** This means that we *should* build on the excellent existing work of our colleagues who have extensively studied the problem of state-based exploration. We would find it not only intellectually dishonest, but *counterproductive* to our investigation if we were to ignore the literature on state-based exploration, and design a fancy novel method for novelty’s sake alone.
>
> The aim of our paper is to conduct a careful and controlled comparison of language vs states. To do this, we need to keep the algorithm constant, and in fact make the *minimal* set of changes required to turn AMIGo into L-AMIGo and NovelD into L-NovelD. We disagree that every paper needs some new method that is as different as possible from existing work. Doing so would confound our results: are our gains due to language, or are they due to our new method? On this point, note that Reviewer auiq, for example, believes it is a **strength** of our paper that our proposed methods are “clear and clean extensions of prior work.”
>
> Finally, we encourage the reviewers and AC to ask: even if it is a straightforward idea to change the input space of AMIGo/NovelD to language, would researchers find the insights of our work useful? We think so, because we found the extension of these algorithms to language highly non-trivial. For example, we designed L-AMIGo so that it does not require knowing the space of possible language goals beforehand; the teacher selects from a dynamically updating buffer of messages the student encounters in the environment. We also propose factorizing the teacher into a grounding network and a policy network, which are trained at separate intervals and with separate objectives. Similarly, in L-NovelD, Appendix E.2 shows that naively adding language into the representations used for the RND network actually hurts performance in MiniHack tasks. And again, outside of algorithmic concerns, we believe it is useful to the broader research community to have empirical results demonstrating the tradeoffs between the design choices in this space.
>
> **Above all,** we hope the reviewers and AC can empathize with the perspective that our paper does not aim to champion a novel and singular exploration method, but rather ask a scientific question—how can (intermediate, noisy, state-based) language improve exploration?—modifying the existing SoTA and running experiments as needed to answer this question.

---

> > ### Author Response · Authors · 2022-08-02
> > **Response to reviewer dbwv (continued)**
> >
> > # Dependence on an oracle language annotator
> >
> > Reviewer dbwv also notes that one potential weakness of the work is that it relies on an oracle language annotator, which limits the applicability of our ideas. While it is true we would like to relax the oracle assumption in the future, we make two points:
> >
> > - Language “oracles” are everywhere in the world. There are a huge number of games and tasks in RL research that come pre-equipped with language. This includes NetHack (which we explored in this setting) but an entire genre of text-based games ([Cote et al., 2018](https://arxiv.org/abs/1806.11532)), and in fact nearly any modern video game, including RL Benchmarks such as StarCraft II ([Vinyals et al., 2017](https://arxiv.org/abs/1708.04782)) and MineCraft ([Guss et al., 2019](https://arxiv.org/abs/1907.13440)). Beyond games, other important RL tasks with language include city/street navigation ([Chen et al., 2018](https://arxiv.org/abs/1811.12354)) and computer control ([Shi et al., 2017](http://proceedings.mlr.press/v70/shi17a/shi17a.pdf)). Hopefully this shows there are a vast variety of tasks of interest to the RL community where our methods are relevant.
> > - Reviewer dbwv presumes that there is no “method to develop language abstractions for environments in general.” **We actually disagree with this**, and argue that such methods are becoming more and more common. For example, [Zeng et al., 2022](https://arxiv.org/abs/2204.00598) use multimodal pretrained models to generate language descriptions of states in an RL framework (but did not examine exploration); [Tarasov et al., 2022](https://openreview.net/forum?id=Spf4TE6NkWq) show that language descriptions can be automatically collected from environments that seemingly don’t have language, including continuous control and finance domains (but again, do not examine exploration). In L90 we also cite [Tam et al., 2022](https://arxiv.org/abs/2204.05080) who show that pretrained models can be used to describe states in photorealistic environments. One could also train a specialized language annotation model offline from (language, state) pairs. One related work, ELLA ([Mirchandani et al., 2021](https://arxiv.org/abs/2103.05825)), does this for MiniGrid, which can be straightforwardly adapted to our setting.
> >
> >   We could have done something similar to these approaches, but this adds substantial additional complexity to our work that is orthogonal to the question of how best to use these language descriptions for exploration. We believe needing to train every part of our problem from scratch is unnecessary and outside the scope of a single conference paper.
> >
> > # Grounding network ablation
> >
> > Yes, we ran that ablation, and the results are in Appendix E.1. This is mentioned in the main text in Footnote 7. The overall results are that including the grounding network does not significantly boost performance (more so for MiniGrid than MiniHack), but *substantially* increases training stability (for example, compare the error bars in the KeyCorridor environments).
> >
> > # Summary
> >
> > We hope our response addresses your concerns, especially surrounding the perceived novelty of our paper. Please let us know if you have any further questions. Given this additional discussion, we hope you consider raising your support for our paper.

---

> ### Author Response · Authors · 2022-08-08
> **Have we addressed your concerns?**
>
> Thanks again for your time and effort in reviewing our paper! As you are the remaining reviewer who is still borderline on the paper, we would like to ask if you have looked at our response, and if so, whether they address your concerns about novelty and applicability? We would be happy to discuss further if you have any remaining concerns. Thank you!

---

### Author Response · Authors · 2022-08-02
**Response to all reviewers**

**Thank you to reviewers for their insightful and helpful reviews, and for the AC for managing this process!** We are glad that reviewers found the paper **“clear and concise”** (dbwv), **“very well written”** (c6cu), **“thorough and well-organized”** (auiq), as well as technically sound: **“I cannot find much to fault with the paper”** (dbwv), **“impressive set of results”** (c6cu).

We have responded to the concerns of each reviewer individually in separate replies.

Overall, it seems like the main concerns voiced by reviewers are related to

1. novelty (reviewer dbwv)
2. a question on the baselines used in the work (reviewer c6cu), and
3. clarification on the contributions of our work (reviewer auiq).

Therefore, we hope that all reviewers and AC especially consider our response to reviewer dbwv on novelty, where we clarify the framing and contributions of our paper, and **explain how building on prior work is a crucial and deliberate part of our contribution.** Second, we believe we have fully addressed reviewer c6cu’s concerns by clarifying the use of language in our baselines and rerunning some of the baselines that did not use language. Our new results, while less dramatic for MiniGrid than they were before, continue to support our main conclusion. Finally, we discuss with reviewer auiq the precise contributions of our work and how it relates to the literature on state abstractions, language, and exploration.

Given these responses, we hope reviewers will consider raising their support for our paper. We are also standing by and willing to address any further questions or concerns, should reviewers have any.

---

### Author Response · Authors · 2022-08-07
**Let us know if we've addressed your concerns**

Thanks to all reviewers again for their time and effort in reviewing our paper. As the discussion period is coming to a close, we would like to know if we have resolved your concerns expressed in the original reviews, and whether you would consider updating your reviews in light of our response. If not, we invite reviewers to express what remaining concerns they have. Thanks again!

---

### Meta-Review · Area_Chair_GGb8 · 2022-08-25

**Recommendation:** Accept
**Confidence:** Certain

**Metareview:**

This paper studies an interesting problem, and overall the reviewers agreed the exposition and validation are sufficient, although there are minor concerns about novelty. However, the work is clear and studies an interesting idea for the RL community. We encourage the authors to consider the issues raised by the reviewers and further improve the work in the final version.


**Award:**

No

---

### Decision · Program_Chairs · 2022-09-14

Accept